# Analysis of COVID-19 Prevention and Control Effects Based on the SEITRD Dynamic Model and Wuhan Epidemic Statistics

**DOI:** 10.3390/ijerph17249309

**Published:** 2020-12-12

**Authors:** Yusheng Zhang, Liang Li, Yuewen Jiang, Biqing Huang

**Affiliations:** 1Department of Automation, Tsinghua University, Beijing 100084, China; zhangyus18@mails.tsinghua.edu.cn (Y.Z.); liang-li17@mails.tsinghua.edu.cn (L.L.); 2Clinical College of Chinese Medicine, Hubei University of Chinese Medicine, Wuhan 430072, China

**Keywords:** dynamics model, control strategy, COVID-19

## Abstract

Since December 2019, millions of people worldwide have been diagnosed with COVID-19, which has caused enormous losses. Given that there are currently no effective treatment or prevention drugs, most countries and regions mainly rely on quarantine and travel restrictions to prevent the spread of the epidemic. How to find proper prevention and treatment methods has been a hot topic of discussion. The key to the problem is to understand when these intervention measures are the best strategies for disease control and how they might affect disease dynamics. In this paper, we build a transmission dynamic model in combination with the transmission characteristics of COVID-19. We thoroughly study the dynamical behavior of the model and analyze how to determine the relevant parameters, and how the parameters influence the transmission process. Furthermore, we subsequently compare the impact of different control strategies on the epidemic, the variables include intervention time, control duration, control intensity, and other model parameters. Finally, we can find a better control method by comparing the results under different schemes and choose the proper preventive control strategy according to the actual epidemic stage and control objectives.

## 1. Introduction

Epidemic disease propagation that involves large populations and wide areas can have a significant impact on society. COVID-19, a new infectious disease, reported in Wuhan, China, in late 2019, and has been reporting around the world. There are more than 80,000 confirmed cases and 4000 death cases in China, and tens of millions of confirmed cases all over the world so far. Understanding transmission dynamics of the epidemic can help us study the transmission process and development trend of the disease [1,2,3]. Different dynamic models were proposed for different diseases and successfully simulated the trends of infectious diseases [4,5]. For reducing the losses caused by the epidemic and limiting the continuous growth of the infected number, it is important and urgent to adopt appropriate control strategies to control the epidemic.

Among all the intervention methods, vaccination is considered to be one of the most useful and cost-effective strategies to control the spread of epidemic disease [6]. Epidemiological models that incorporate the impact of vaccination [7] can be useful to determine the effective ways of controlling the spread of disease. Moreover, there are researches that focus on the optimal vaccine distribution when we have limited vaccine resources [8,9,10,11]. During the COVID-19 outbreak, however, we face more difficult control conditions, in which is there is no specific medicine and no vaccine. Therefore, methods other than drug control should be adopted for intervention, such as social distancing and travel restrictions.

Shim [12] considered the effect of social distance on disease transmission control. Social distancing can control the epidemic at a lower cost, and this method was indeed widely used during the period of the new coronavirus and achieved good results. The World Health Organization (WHO) has organized a campaign called SAVE LIVES: Clean Your Hands every year [13], which aims to bring people together in support of hand hygiene improvement globally. According to the research in [14], if the handwashing rates of airport passengers are raised, the disease-spreading speed can be reduced by 69% at most. The research focuses more on places with high mobility, such as airports. However, the proportion of people traveling by air during the epidemic period is relatively low, so the corresponding prevention and control methods should be studied for large-scale regions and universal travel schemes. Many other studies on trend prediction and control methods under special conditions during the COVID-19 period have also been carried out. Leung [15] predicted the impact of the Wuhan outbreak on the global epidemic. Their research focused on the effect of transportation in the spread of the epidemic and was based mainly on traffic data. Zhao [16] estimated the basic reproduction number of COVID-19 in China at the beginning of the epidemic and found that the early outbreak largely follows an exponential growth pattern. Zandavi [17] forecasted the spread of COVID-19 under control scenarios using LSTM (Long Short-Term Memory) and dynamic behavioral models. He used time series models to predict the trend of the epidemic and concluded that skilled medical staff and high-quality hospitals would be able to control the outbreak. Giordano [18] modeled the COVID-19 epidemic and interventions in Italy. They predicted the trend of the epidemic in Italy and analyzed the impact of various policies on the epidemic. Anderson [19] and his group estimated the impact of COVID-19 control methods. Their studies classified the patients into 12 categories according to their medical condition and severity. Then they compared the trend of the epidemic with different levels of control. These researches can give decision-makers more inspiration for improvement directions. It also has more guiding significance for the prevention and control of the next epidemic.

Although a variety of measures and conditions were adopted during the COVID 19 epidemic, more complex schemes should be considered. The same control method will produce different control effects with different control intensities at different times. The main objective of this paper is to observe and compare the differences of control effects brought about by these different control scenarios. And the control strategy most suitable for the current control scenario will be obtained under the given conditions and control target.

The organization of this paper is as follows. An epidemic disease SEITRD model is proposed and we discuss the propagation process of the model in Section 2. In Section 3, we present the prevention and control measures taken during the COVID-19 outbreak and how they behaved in the model. Moreover, we conducted simulations based on different prevention and control strategies in Section 4. Finally, we present the discussions and conclusions in Section 5 and Section 6.

## 2. Materials and Methods

### 2.1. SEITRD Model

For diseases with the incubation period, the SEIR model is generally used for simulating the transmission process of diseases [20]. The basic SEIR model divides the population into four categories: S (susceptible), E (exposed), I (infection), and R (recovered). However, according to the characteristics of different diseases and the macro-control measures to deal with these diseases, adjustments should be made for realistic situations. Lu Guo [21] added stage T (treatment) to represent patients receiving hospital treatment and assistance, which indicates the effect of medical treatment on disease inhibition. In their model, there was no distinction as to whether or not a person in stage E sought medical attention. On the basis of the SEIR model, Anderson [19] and his group considered the effect of self-isolation and divided each stage into isolated and un-isolated parts for separate analysis. Based on the above two models, we considered the realistic situation that patients in stage E would actively seek medical treatment during COVID-19, and added the T stage in the model. The transformations between other stages and the T stage were also included. Besides which, the data of death is easy to acquire, so we added a D stage for corresponding representation. Figure 1 shows the transmission dynamics of the SEITRD model. The model adds two categories based on the SEIR model: T (treatment) and D (death). In view of the effect of self-isolation and the changing epidemic response, we adopted the variational transmission probability. For COVID-19, there is a difference between the adults, the children, and the elderly. However, we focus on the population of the whole city, which has no individual attributes. At the same time, the measures taken during the outbreak are also nationwide, and there was no special policy or arrangement for a certain group of people. Therefore, our model only classifies the population by the disease stage.

The propagation model is based on the following assumptions.

Everyone is at risk except those in R stage and D stage.People in E stage cannot be directly converted to I stage, and a certain incubation period is required.People in E stage and patients in I stage have the same infectious capacity.Patients in T stage are not infectious under the protection of the hospital.People in R stage do not get sick again.

Based on the above assumptions, the propagation dynamics equations of the model are shown as follows.
(1)$$\frac{dS}{dt}=-\alpha E\frac{S}{N}-\alpha I\frac{S}{N}$$
(2)$$\frac{dE}{dt}=\alpha E\frac{S}{N}+\alpha I\frac{S}{N}-pE\gamma_{1}-\left(1-p\right)E\gamma_{2}$$
(3)$$\frac{dI}{dt}=pE\gamma_{1}-I\theta -I\beta_{1}-I\varepsilon$$
(4)$$\frac{dT}{dt}=\left(1-p\right)E\gamma_{2}+I\varepsilon -T\theta -T\beta_{2}$$
(5)$$\frac{dR}{dt}=\;T\beta_{2}+\;I\beta_{1}$$
(6)$$\frac{dD}{dt}=\;T\theta +\;I\theta$$

The specific meanings of the identifiers can be found in Table 1. Furthermore, we used the population of Wuhan to represent the total population in the model. We didn’t consider the natural birth rate and natural death rate during an outbreak, or the migration into and out of Wuhan during the outbreak. The dynamics equation of population can be written as Equation (7), where *N* is the total population.
(7)$$\frac{dN}{dt}=-\;T\theta -\;I\theta$$

### 2.2. Epidemic Parameter

Choosing proper parameters and accurate status data is vital for simulating the epidemic trends. We referenced the work of Lu Guo [21] to set the majority of the variables and parameters and add some new ones to meet our requirements. We processed the epidemic data published by the national health commission and analyzed the transmission characteristics of COVID-19. The final values of variables and parameters of the SEITRD model are shown in Table 1.

The variables include S, E, I, T, R, D, which equal the number of people in the different stages. The values of the variables in the table are the initial numbers at the start of simulations. 21 January was set as the initial date for the outbreak in our experiments. *N* equals total population in Wuhan from the public data and the initial value of S is estimated by *N*. Considering the delay before infection medical treatment, we use the number of confirmed cases three days after the starting date as the initial number in stage I, and the number of confirmed cases five days after the starting date as the initial number in stage E. The estimated values of E and I may be different from the actual situation, and we modified these values according to the fitted values of 15 days after the starting date, which means that the values in Table 1 are the corrected values. The values of T, R, and D are from the actual data of the Chinese Center for Disease Control and Prevention (CDC). Public data and CDC data can be found at the National Health Commission, People’s Republic of China [22]; or the Chinese Center for Disease Control and Prevention [23].

The other rows in Table 1 are parameters of the model. We used a similar approach like E and I to estimate parameter α, which was also modified after a short-term simulation. Parameter *p* is the hospitalization ratio of the exposed population. In our model, the exposed population may get medical treatment before getting into stage I, or directly get into stage I after the incubation period. The value of *p* should be between 0 and 1 and we take the average value of 0.5 as its initial value, which will vary in different experimental scenarios. The death rate θ is determined on the basis of the proportional relationship between the number of deaths and the number of confirmed cases. $\beta_{2}$ is based on the time from medical treatment to recovery, which is usually seven days, so the parameter was set as the reciprocal of the interval. $\beta_{1}$ means that the recovery probability of those who have not sought medical treatment is lower than that of those who have sought medical treatment, which is assumed to be 0.1. Similarly, the transformation among the three stages of E, I, and T also takes time because there’s an incubation period [24] before developing symptoms and there is also a certain delay before getting medical treatment. In our model, we use ε, $\gamma_{1}$, and $\gamma_{2}$ to represent these delay effects, estimated by the reciprocal of the delay. The usual incubation period for the disease is seven days, so $\gamma_{1}$ is 0.15. The active treatment event in the stage E population should be within 1–7 days, so we take 0.3 as $\gamma_{2}$’s assumed value. For the people in the I stage, it takes some time (two days in this paper) to go to the hospital after having symptoms, we use 0.5 to represent the effect.

### 2.3. Model Simulation and Trend Prediction

In the early stage of the COVID-19 outbreak, there were many studies on trend prediction and prevention and control methods based on the classic SEIR model. Tang [25] and his group considered the effect of the isolation mechanism on the transmission process and made a distinction between symptomatic and asymptomatic infected persons. But they think too carefully about some stages, and their predictions are relatively high. Leung [15] used the SEIR model completely in their prediction, mainly considering the impact of transportation factors on disease transmission. But they did enough to analyze the isolation mechanism and medical intervention. Inadequate consideration of prevention and control options also overstates their predictions. Chen [26] took into account the time-lag of interpopulation transfer and the impact of quarantine measures on the development of the epidemic. However, in terms of prevention and control methods, they overemphasized self-isolation, a costly prevention and control method, and did not explore more other prevention and control methods. Moreover, their lack of information on the classification of patients who did not develop the disease and the medical treatment activities of the general public affected the accuracy of their predictions.

Based on the preset model transmission parameters and initial state, the expected trend of the epidemic is shown in Figure 2. It shows that the percentage of people in stage E peaked in early February. The peaks for stages I and T are in mid-February and the overall outbreak is largely contained in April or May. We have also simulated the outbreak in Egypt, and the results are in Appendix A.

Due to the lack of early statistical data, delayed statistical data, and people’s lack of awareness, it is very difficult to fully and accurately reconstruct the transmission process of the disease. The parameters in the model vary irregularly during the spread of the disease. Therefore, in the area of trend prediction, this paper expects to restore a dynamic model with similar trends and relatively consistent parameters, rather than accurate trend results.

Even now, there are some small recurring outbreaks that cannot be considered as part of the outbreak because of the small number of confirmed cases. These recent small-scale outbreaks should be simulated and studied as a new outbreak, which are not within the predicted range of this study.

### 2.4. Evaluation Methods for Disease Control

The impact of the epidemic is multifaceted, including the impact on people’s normal lives and the stagnation of economic development. Some previous studies [27,28] have also quantified the effects as economic measures to determine the optimal solution. These studies generally measure the medical input and the loss of patients due to the disease with the economic dimension, and the specific quantitative method should be treated separately according to the respective characteristics of the disease.

In the COVID-19 epidemic, however, the Chinese government has spent much to help Wuhan through a hard time. And the purpose is to reduce the number of people affected by the epidemic. Therefore, we use the impact of the epidemic on people to measure the loss caused by the epidemic. For the same patient, being ill for twenty days will affect him more than being ill for ten. So, we consider the cumulative duration of illness as a measure of epidemic loss. Similarly, patients in stage E may be affected less than the patients in stage I. Therefore, we represent the impact of the epidemic by a weighted cumulative number of affected days for each stage of the patients. The formula is as Equation (8).
(8)$$J=\int_{0}^{t_{f}}(\lambda_{1}E_{t}+\lambda_{2}I_{t}+\lambda_{3}T_{t})dt$$

In the formula, $E_{t}$, $I_{t}$, $T_{t}$ represents the number of patients in stage *E*, stage *I*, and stage *T* at time t. $\lambda_{1}$, $\lambda_{2},\lambda_{3}$ are unequal equilibrium parameters of the three stages. They can be assigned different values depending on the degree of influence. In our experiments, given the specificity of COVID-19, the parameters are all set to 1. $t_{f}$ is the 60th day after 21th January, which is the final time of our experiments.

## 3. Control Strategies

### 3.1. Travel Restrictions and Other Self-Control Methods

Immunization and isolation are the two main ways to control an outbreak [29]. In the absence of a corresponding treatment drug and vaccine, isolation is the primary means by which countries contain outbreaks. Isolation is divided into isolation of patients and isolation of susceptible groups. Travel restriction is a control measure that isolates susceptible people. During the epidemic, the Chinese government has taken a series of measures [30] to prevent and control the outbreak, such as sealing off cities, putting travel bans in place, and cutting down parties. These measures are mainly aimed at reducing people’s contact with each other to minimize the probability of infection. There are also personal-level precautions, such as frequent handwashing and wearing masks, that also reduces the risk of infection during exposure.

A reduced contact rate between people leads to a reduction in infection rates in the model, which means that the parameter α changes over time. In most propagation dynamics models, α is a constant number. In this paper, we use the infection rate to reflect the government control level. With the outbreak of the epidemic, the level of prevention and control showed an upward trend in the early stage. This parameter variation trend corresponds to the rapid response of the public in the early stage of the epidemic and smooth implementation of travel restriction in the later stage. Subsequent experiments take this parameter set as a benchmark for analysis and comparison.

### 3.2. Medical Testing and Other Hospital-Control Methods

Timely medical testing and treatment is an effective means in the process of infectious disease control [31]. In the course of the COVID-19 epidemic, due to people’s lack of awareness and the shortage of medical resources, not all the infected people will choose to seek medical treatment, and the exposed persons do not often seek medical testing immediately because their symptoms are not obvious. Although hospitalization does not result in an immediate recovery, doctors can manage the patients in a way that reduces the risk of disease progression and transmission to others. It’s also a way of isolating the patients. Therefore, it is very important to evaluate and measure the percentage of patients in Wuhan accurately to judge the trend of the epidemic. The relevant parameter in the model is *p*.

In this paper, we use a piecewise linear function to determine the parameter *p*, and the formula is as Equation (9).
(9)$$p_{t}=\{\begin{array}{l} \;\;\;\;p_{1},\;\;\;\;\;\;\;\;\;\;\;\;0\leq t\leq t_{1}\\ \;\;\;\;p_{2},\;\;\;\;\;\;\;\;\;\;\;t_{1}\leq t\leq t_{2}\\ \;\;\;\;p_{3},\;\;\;\;\;\;\;\;\;\;\;t_{2}\leq t\leq t_{f} \end{array}$$
$t_{1}$ and $t_{2}$ represent the key time nodes of government policies corresponding to the time of Wuhan city closure and the completion time of the new hospitals. At these two time points, due to the increase of public awareness and the guarantee of medical equipment, the percentages of hospitals in stage E and stage I will increase. The parameter $t_{f}$ represents the final time of the outbreak, which is around the 60th day after the outbreak in our model.

## 4. Simulations

Based on the analysis of the various states and the changes of the epidemic, the trend of the epidemic and the changes in the number of people in each stage have been predicted. In this section, we will compare and analyze the effects of preventions and control measures with different intensities. Although there is population heterogeneity in the spread of disease, we believe that such characteristics are distributed relatively evenly, that is, we can use an average population attribute to represent the population characteristics of the region.

We correct the parameters of the model by analyzing the characteristics of the epidemic, and this section mainly compares the effects of various prevention and control methods. We mainly compared the differences in the development of the epidemic caused by three prevention and control variables, including hospitalization ratio, control time, and control interval. When analyzing and simulating each control variable, we keep the other variables consistent so that we can compare and analyze them. For example, when we analyzed the differences in prevention and control effects caused by hospitalization ratio, we kept other parameters consistent with those in Table 1 to observe the influence of a single variable on the results.

### 4.1. Simulations of Different Hospitalization Ratios

We compared the trends of the epidemic under different hospitalization ratios. Figure 3 shows the comparison of percentages of people in five stages at different hospitalization ratios. The corresponding hospitalization ratios of the five curves are shown in Table 2. The three hospitalization ratios correspond to three different values of parameter *p*, described in Section 3.2. Scheme 1 was treated as the baseline in which the hospitalization ratio is 0.5. In scheme 2 and scheme 3, the hospitalization ratios in three periods are increased or reduced together. In scheme 4 and scheme 5, hospitalization ratios decreased in the early period and increased in the later period. The key time nodes $t_{1}$ and $t_{2}$ are the 20th and 40th days.

In Figure 3a–c, we can observe the influence of increasing the overall hospitalization ratio on the number of patients at each stage. As the overall hospitalization ratio rises from 0.4 to 0.6, there is an overall decrease in the number of patients in stage E and I. The peak number of patients in stage E changes from nearly 500,000 to less than 400,000. However, the number of people in stage T showed an increasing trend, caused by the increased percentage of patients seeking medical treatment, which has a positive impact on the development of the epidemic.

In Figure 3a,d,e, while the overall hospitalization ratios are roughly the same, we adjusted the hospitalization ratio in the early stage and later period of the epidemic. It is close to the reality of trends of medical treatment: looseness in the front and tightness in the back, or tightness in the front and looseness in the back. By comparison, it can be found that the peak of stage E in Figure 3e is the highest among the three schemes, and the peak of stage E in Figure 3d is also higher than that in Figure 3a. In other words, a higher hospitalization ratio in the early stage can effectively reduce the impact of the epidemic. From Figure 3 and Table 2, we also find that a higher hospitalization ratio cannot fully make up for the loss caused by the lower hospitalization ratio in the early stage.

By comparing the total affected days of the five schemes, we can reach three conclusions as follows:A higher hospitalization ratio could significantly reduce the impact of the epidemic by comparing scheme 1, scheme 2, and scheme 3. Increasing the hospitalization ratio by one-tenth could reduce the impact of the epidemic by one-tenth correspondingly.By comparing scheme 1, scheme 4, and scheme 5, we can find that a higher hospitalization ratio in the early stage of the disease will limit the spread of the disease more obviously.By analyzing the above results and the overall trend of the outbreak, measures to increase the hospitalization ratio are very effective. Seeking medical testing and treatment can speed up recovery and reduce the chance of infecting others, which is reflected in the model in that the recovery rate is increased and the infection rate is reduced.

### 4.2. Simulations of Different Intervention Times

The Chinese government promptly took prevention and control measures shortly after the outbreak so that the outbreak could be effectively controlled in a short time. However, there was still a delay between the first group of infection cases and the government’s response. In addition, it takes time to evaluate the severity of the epidemic and build new hospitals. If these measures were taken forward or delayed for a few days, how would the epidemic trend change? In this section, we change the contact rate at $t_{1}$ and $t_{2}$, and the parameter drops from 0.6 to 0.4. The time point of change is set to be different for comparing the impact of the application time of measures on disease control.

As shown in Figure 4, the change trend of the three stages E, I, and T is roughly the same, and the percentage or relationship among them will not change with the advance or delay of the prevention and control time. The intervention time and total affected number corresponding to each scheme are shown in Table 3. Scheme 1 is the baseline. $t_{1}$ is earlier in scheme 2, scheme 3, and later in scheme 4, and scheme 5. $t_{2}$ is earlier in scheme 3 and later in scheme 5. It is obvious that the measure in scheme 3 is the strictest, while the measure in scheme 5 is the most relaxing.

With the advance of the epidemic prevention and control time, the peak of the epidemic will be relatively delayed, and the peak of the number of patients will be relatively reduced. In other words, earlier prevention and control measures can limit both the scope of the epidemic and its speed of development.

By comparing the results in Table 3, it can be found that earlier implementation of prevention and control measures can reduce the impacts of the epidemic to some degree:Prevention and control in advance in the early stage of the epidemic is better than that in the later stage by comparing scheme 1, scheme 3, and scheme 5.Early interventions could reduce the overall impact of the epidemic and delay the peak of the epidemic at the same time.When the intensity of prevention and control is low, increasing people’s awareness of seeking medical treatment may be more effective than informing people to reduce their exposure by comparing Table 2 and Table 3.

### 4.3. Simulations of Different Control Intervals

In some cases, we are not able to maintain a high level of containment all the time. A long interval will give enough time for the outbreak to develop, while a short interval will affect people’s daily lives. Interval control can find the balance between epidemic control and daily life. To study the impact of different control intervals, we conducted the following comparative experiments. In the experiments, the control interval was set to five days, three days, and one day respectively. If the control interval is one day, it means that people have the freedom of one day every two days. The shorter the control interval is, the stricter the measure is. The schemes are in Table 4.

The experimental results are in Figure 5. In the experiments, the change range of control intensity is the same, and only the control interval is changed. We find that a smaller control interval results in a better control effect.
A small control interval can reduce the peak number of people affected. In Figure 5a,c, the peak of people in stage E is about 2%, while in Figure 5b, the peak in stage E is about 1.6%.A small control interval delays the arrival of the peak time, which can help slow the growth of the epidemic. In Figure 5a,c, the peak time is around the 25th day, while in Figure 5b, the peak time is around the 30th day.Through the small-interval control scheme, the total affected days has also been reduced. As shown in Table 4, the total affected days in scheme 1 decreased by about 4 percent compared to scheme 2.

Figure 5d compared the trends of the percentage of stage T under the three schemes, and it proved the above conclusions more obviously. In the actual applications of measures, the Chinese government has also limited continuous outgoing, which corresponds to the interval control in the experiment. The experiment in this section also proves the rationality and effectiveness of this measure.

## 5. Discussion

The spread of infectious diseases affects people’s daily life in many aspects, so it’s of great significance to understand the transmission process of the disease, predict the development trend of the disease and take effective control measures. In this paper, we combined the data of confirmed COVID-19 cases to construct the SEITRD model for trend simulation. Then, the influence of control intensity, control time, and control interval on the development of the epidemic was considered comprehensively.

In terms of trend simulation, we used the CDC data and the estimated parameters to predict the epidemic trend. By comparing the predicted number of confirmed cases with the actual number of confirmed cases, we verified the accuracy and rationality of the constructed model. There are still some limitations and deficiencies in the process of establishing the prediction model, which led to differences between the predicted results and the actual results. Firstly, at the model level, as we mentioned in Section 2, many other researchers use the SEIR model or other models based on the SEIR model for trend prediction. Depending on the focus of the study, their models produced different results. Some studies predict the future trend of COVID-19 [16,26], some analyze how to effectively control the epidemic [17,19] and some analyze the trend of the spread among regions [18]. Our paper focuses on the changes of prevention and control measures and modifies the model for specific prevention and control measures, which may lead to inaccurate prediction results. Secondly, we didn’t consider the impact of traffic between cities on the spread of the epidemic. Wuhan is the starting point of the epidemic, and the impact on other cities is within a controllable range. At the same time, under the effect of city closure measures, the effect of inter-city communication may be far less than that of intra-city communication. The studies that take transportation into account are more concerned with the impact of the epidemic in Wuhan on other cities [15]. Finally, the determination of partial parameters mainly depends on the data of confirmed cases and cured cases, while more parameters depend on estimation and calculation, so the accuracy of parameters needs to be further improved.

Based on the trend simulation, we explored the influence of control strategies. The considered strategy variables are divided into control intensity and control time. However, in the actual situation, we would face more diverse and complex control scenarios, such as frequent handwashing and mask-wearing. Besides which, the age and health status of the population also play a role in the spread of the epidemic, and people with poor health status may be more susceptible. In the follow-up study, the influence of some comprehensive factors should be considered to restore the actual control strategy more realistically.

## 6. Conclusions

In this paper, we established the SEITRD propagation dynamic model according to the epidemic characteristics of COVID-19. We determined the parameters of the model according to the real-time epidemic status in Wuhan. Then, the epidemic trend of the epidemic situation in Wuhan was simulated and predicted with the established model. By comparing the real epidemic trend with the simulation results of the model, we found that the simulated epidemic trends were similar to the real epidemic trend, which verified the rationality of our model. The results showed that the outbreak in Wuhan peaked in mid-February and was effectively controlled in late May, which was exactly what had happened in the real world.

During the development of the epidemic, the Chinese government took active prevention and control measures to control the epidemic for the reason that there were no specific drugs and vaccines. The two main prevention and control measures that could be applied were isolating the susceptible people and treating the infected patients. In this paper, the control effects of different measures are compared.

Firstly, in terms of control intensity, the increase of hospitalization ratio can significantly reduce the number of patients who are capable of infection, so increasing the hospitalization ratio and medical treatment rate at an early stage of the epidemic can have a better effect on the control of the epidemic.

Secondly, it is intuitional that earlier intervention could reduce the impact of an outbreak, which is verified numerically in our experiments. Earlier intervention controls can limit the number of people with the disease. In extreme cases, when the first patient is isolated and cured promptly, there is no widespread outbreak. As a result, the government should take early interventions to control the disease.

Finally, interval control is also an effective control scheme, which can help reduce the influence caused by the control measures. Our experimental results show that choosing a proper short-control interval rather than full-time control can effectively control the epidemic situation.

Although we investigated many factors of control measures, it is still hard to keep totally consistent with the actual situation. In the future study, the influence of comprehensive factors should be considered to restore the actual control strategy.

It usually takes lots of time for the government to deploy the control strategies and for people to raise their awareness. Even with the awareness of prevention, how to balance the normal life and control schemes is a difficult problem. What the government needs to do is to identify the outbreak early, respond positively, and decide on a more appropriate control strategy based on the actual situation.

## Figures and Tables

**Figure 1 ijerph-17-09309-f001:**
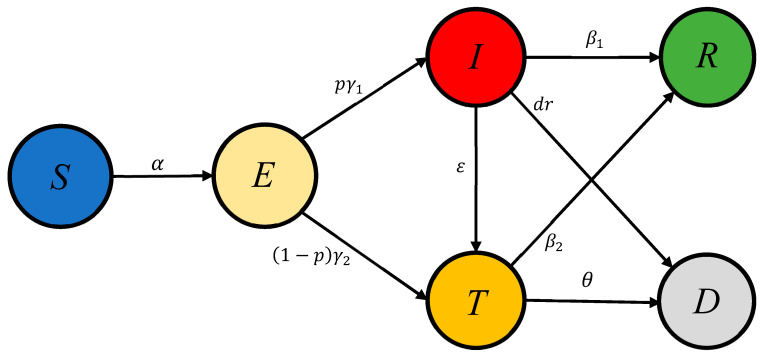
Schematic of the SEITRD propagation dynamics model.

**Figure 2 ijerph-17-09309-f002:**
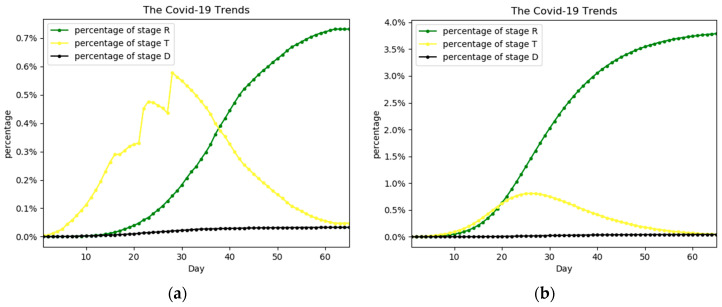
The percentage trend of people in each stage. The actual trend is represented by (**a**), while (**b**) is the predicted trend. The percentage is the percentage of the whole population.

**Figure 3 ijerph-17-09309-f003:**
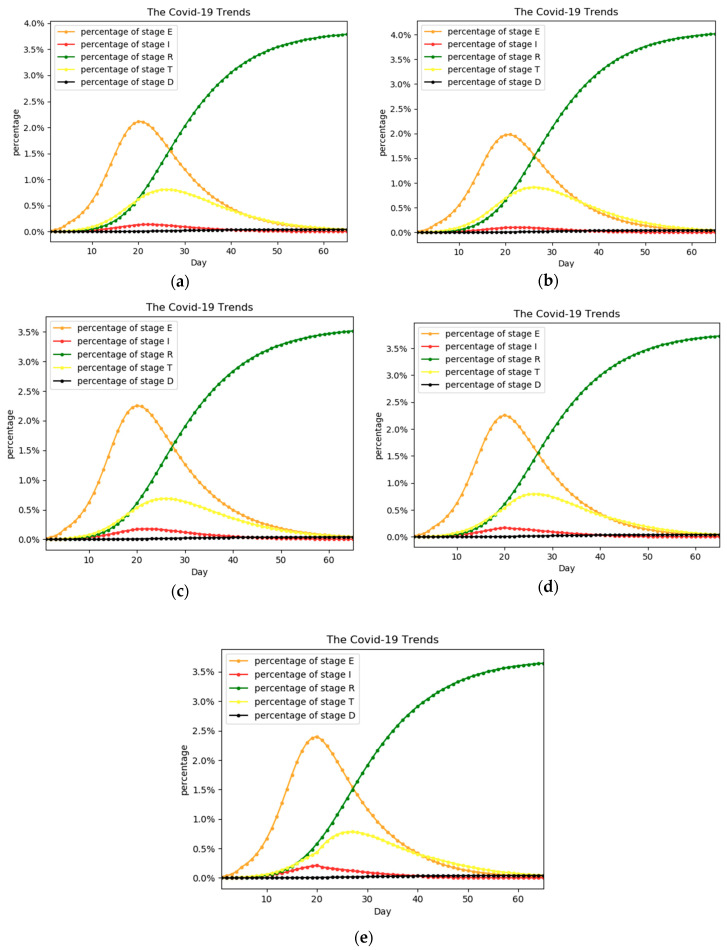
The percentage trends of the five stages under different hospitalization ratios. Figures (**a**) through (**e**) correspond to scenarios 1 through 5 in Table 2 respectively. The overall number of patients in scheme 3 is significantly higher than that of the other groups due to the significantly lower hospitalization ratio in scheme 3.

**Figure 4 ijerph-17-09309-f004:**
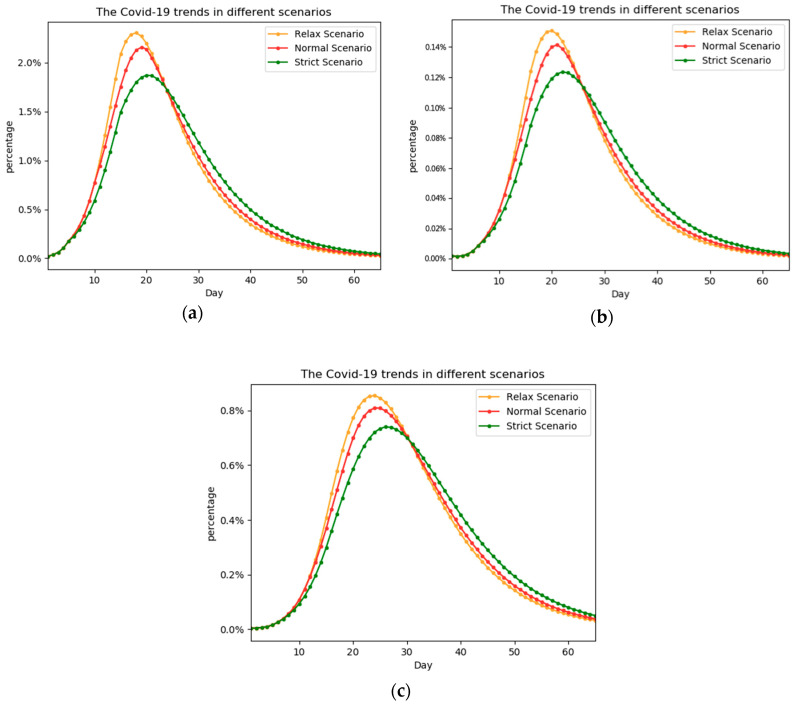
The percentage trends of the E, I, and T stages under different intervention times. Graph (**a**), compares the percentage of people in stage E of scheme 1, scheme 3, and scheme 5. Graph (**b**) compares the percentage of people in stage I. Graph (**c**) compares the percentage of people in stage T. Taking control earlier means stricter control schemes, and strict control schemes can achieve better results in the end.

**Figure 5 ijerph-17-09309-f005:**
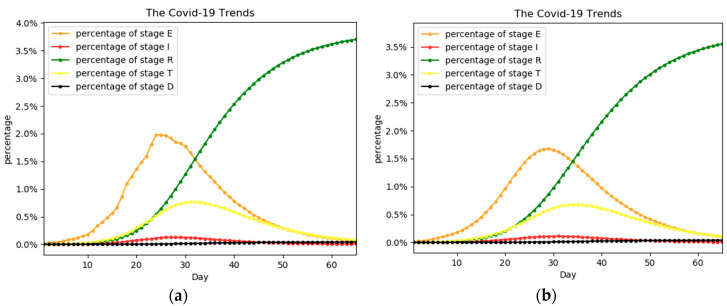
The epidemic trends of each stage under different control intervals. Graphs (**a**–**c**) reflect the overall trend of the epidemic at different control intervals, the control intervals were 1 day, 3 days and 5 days, respectively. Graph (**d**) shows the trend comparison of the percentage of treatment cases under the three schemes. A one-day control interval can be found to have a significant control advantage.

**Table 1 ijerph-17-09309-t001:** Variables and parameters of the epidemic model.

Variable	Explanation	Value	Source
S	susceptible population	≈11,000,000	estimated
E	exposed population	1000	estimated
I	infective population	350	estimated
T	population in treatment	≈10	CDC data
R	recovered population	≈5	CDC data
D	death population	≈3	CDC data
*n*	total population	11,000,000	public data
α	infection rate	0.35	estimated
*p*	hospitalization ratio	0.5	assumed
θ	death rate	1.78 × 10^−4^	calculated
$\beta_{1}$	recovery rate of stage I	0.10	assumed
$\beta_{2}$	recovery rate of stage T	0.15	calculated
ε	frequency of stage I getting into stage T	0.5	assumed
$\gamma_{1}$	frequency of stage E getting into stage I	0.15	estimated
$\gamma_{2}$	frequency of stage E getting into stage T	0.3	assumed

**Table 2 ijerph-17-09309-t002:** Total affected days under different hospitalization ratios.

Schemes	Hospitalization Ratio1 $\left(p_{1}\right)$	Hospitalization Ratio2 $\left(p_{2}\right)$	Hospitalization Ratio3 $\left(p_{3}\right)$	*J* (Total Affected Days)	Increase/Decrease
scheme 1	0.5	0.5	0.5	9,672,234	baseline
scheme 2	0.6	0.6	0.6	8,912,171	−7.9%
scheme 3	0.4	0.4	0.4	10,518,962	+8.6%
scheme 4	0.4	0.5	0.6	9,819,022	+1.5%
scheme 5	0.3	0.5	0.7	10,009,594	+3.5%

**Table 3 ijerph-17-09309-t003:** Total affected days under different intervention times.

Schemes	$t_{1}$ (Day)	$t_{2}$ (Day)	*J* (Total Affected Days)	Increase/Decrease
scheme 1 (normal)	10	20	9,517,831	baseline
scheme 2	5	20	9,472,265	−0.5%
scheme 3 (strict)	5	15	9,381,592	−1.5%
scheme 4	15	20	9,577,645	+0.6%
scheme 5 (relaxed)	15	25	9,646,198	+1.3%

**Table 4 ijerph-17-09309-t004:** Total affected days under different control intervals.

Schemes	Interval	*J* (Total Affected Days)	Increase/Decrease
scheme 1(strict)	1	9,099,989	−4.0%
scheme 2(normal)	3	9,483,481	baseline
scheme 3(relax)	5	9,555,856	+0.8%

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
