# Peer review of "Analysis of COVID-19 Prevention and Control Effects Based on the SEITRD Dynamic Model and Wuhan Epidemic Statistics"

_ijerph, 2020, doi:10.3390/ijerph17249309_

Round 1
Reviewer 1 Report
Dear authors,
I am glad I have this opportunity to review your research. It is dealing with the current topic associated with COVID-19 (which is very actual). Honestly, I do not know if this topic is not overcrowded yet (but I am not a person for its evaluation). Anyway, my comments are the following ones:
- modify the title by adding demographics aspect (local, international, world, ...);
- add and specify your main goal/aim before the manuscript structure (line 60, page 2);
- support the SEITRD Model (Sub-section 2.1) by any citations, add authors of this model and present its previous usages;
- locate the letters in bubbles (Figure 1) in the middle;
- how did you estimate 4 variables from Table 1? (e.g. infective population). I consider the explanation "Exposed population and infective population were extrapolated from subsequent epidemic trends" not sufficient;
- are variables from Table 1 used by any author or this combination is your own? If they are taken over, please explain your novelty. If they are your own, please a selection procedure. Each of these possibilities supports by citations;
- I am on page 6 and I still do not know what is your goal/aim, please add and specify it.
- consider any structure modification (sub-section are too short, e.g. 3.2);
- you are mentioning a lot of information without any citation (e.g. sub-section 3.1). I am not sure that each of this information is your own one;
- each scheme should be better described and explained (Figure 3), this figure could be also better prepared (gaps are too big), the same for scenarios in Figure 4;
- check the formal processing of your manuscript (Figure 4 is twice);
- Discussion should be realized in the context of the results of other authors, which is totally missing. Please rewrite this section.
I wish you all the best with this manuscript and other ones in the future. I also hope my comments could help you to improve its quality and scientific soundness.
Reviewer 2 Report
Overview
This is a manuscript that looks at a new SEITRD dynamic epidemiologic model, similar to the widely used SEIR model for COVID transmission. The authors then utilize the model to compare to empiric data from Wuhan.
Major
There are significant grammatical errors in the manuscript. Please review with a native English speaker and revise prior to further review.
The readers would benefit from a comparison of the SEITRD dynamic model compared to the widely used SEIR models as these are generally considered the gold standard. While comparing empiric outcomes is important there is the possibility of creating the model after the fact, and is important to compare performance to other more broadly used modeling schemas.
The readers would benefit from predictive analysis on other locations to ensure this model is more generalizable beyond Wuhan
Minor
Lots of grammatical errors and syntax optimizations--too many to go through. Please review with a native english speaker
10 which has caused enormous losses.
12 Proper prevention and treatment methods has been a hot topic of discussion.
16 “through dynamical behavior model, remove “of the”
60 “chapter” is used, these are rather sections.
Reviewer 3 Report
In this article the authors use an epidemic compartmental model and through simulations examine the impact of intervention time, control duration, and control intensity on the expansion of COVID-19 in Wuhan. The idea of this study is clear in the paper, but I have few comments and suggestions that I would like to ask the authors to consider. My comments are the following:
- In my opinion, the biggest limitation of your study is that you consider a homogeneous (well-mixed) population where each susceptible individual has the same probability to get the infection at each time step. However, in reality the structure of each population is complex, and the transmission rate depends on how many contacts each individual has. This is the reason why individuals with many contacts are considered as hubs and are super-spreaders. Can you please comment on that?
- On page 1, you state that “Vaccination is considered to be the most useful and cost-effective strategy to control the spread of epidemic disease”. However, in the absence of vaccines for new viruses such as the coronavirus, the WHO considered hand-hygiene as the first prevention step against the epidemic expansion. A recent study has also shown that an increase in handwashing rate by travelers can reduce an epidemic expansion by up to 69%. Can you please comment on that?
- Can you please give more information on how the values of the “estimated” and “calculated” parameters in table 1 have been derived? For example, you state that the estimation of beta1 and beta2 is based on the recovery time, but you don’t explain their relationship.
- Can you please add in section 3.1 some citations for other studies that examined the effect of travel restrictions during COVID-19?
- On page 6, you state that “Many previous studies have also quantified the effects as 174 economic measures to determine the optimal solution”. Can you please add some citations?
- Can you please provide some more information for the simulations? For example: Are you using agent-based simulations? Do you apply Monte-Carlo simulations? How many randomizations have you done for each simulation scenario?
- Can you please add confidence intervals around the epidemic curves in Figures 3-5?
- In my opinion the vertical axes of Figures 2-5 should indicate the fractions of each compartment in the population and not the actual numbers as using fractions it is easier to interpret the differences between the different scenarios and variations of parameters.
Round 2
Reviewer 2 Report
The authors have considerably improved the English grammar with significant revisions
SEIRTD model now contrasts with the widely used SEIR model which is helpful to the reader
All major and minor comments have been revised. Thank you.
Author Response
Thanks again for the helpful comments. Thank you for the considerable time and effort to review our work.
Reviewer 3 Report
Thanks for replying to my comments. I have few new minor comments based on your changes:
- Citation 13 is not authored by WHO. For WHO see for example the "SAVE LIVES: Clean Your Hands" campaign.
- Please use proportions also in the vertical axes of Figure A1.
- Because there is a difference between proportion and percentage please use one of those. It is confusing to have the label of "proportion" in the y-axis of the graphs and the symbol of "%" in the y-axis numbers.
